# Affective Responses during High-Intensity Interval Exercise Compared with Moderate-Intensity Continuous Exercise in Inactive Women

**DOI:** 10.3390/ijerph18105393

**Published:** 2021-05-18

**Authors:** I-Hua Chu, Pei-Tzu Wu, Wen-Lan Wu, Hsiang-Chi Yu, Tzu-Cheng Yu, Yu-Kai Chang

**Affiliations:** 1Department of Sports Medicine, Kaohsiung Medical University, Kaohsiung 807, Taiwan; wenlanwu@kmu.edu.tw (W.-L.W.); u97009049@gmail.com (H.-C.Y.); u107805002@kmu.edu.tw (T.-C.Y.); 2Ph.D. Program in Biomedical Engineering, College of Medicine, Kaohsiung Medical University, Kaohsiung 807, Taiwan; 3Department of Medical Research, Kaohsiung Medical University Hospital, Kaohsiung 807, Taiwan; 4School of Physical Therapy & Athletic Training, Pacific University, Hillsboro, OR 97123, USA; wuel@ohsu.edu; 5Department of Physical Education, National Taiwan Normal University, Taipei 106209, Taiwan; yukaichang@ntnu.edu.tw; 6Institute for Research Excellence in Learning Science, National Taiwan Normal University, Taipei 106209, Taiwan

**Keywords:** inactive women, affect, psychological responses, high-intensity interval exercise

## Abstract

This study aimed to investigate the effects of an acute bout of high-intensity interval exercise (HIIE) and moderate-intensity continuous exercise (MICE) on affective responses in inactive women. Thirty women with normal body mass index (BMI) and 30 women with BMI ≥ 24 kg/m^2^ participated in the study. All participants completed a graded exercise test and performed two exercise sessions (HIIE and MICE) in random order. Affective responses were assessed during and after each exercise session, using the rating of perceived exertion (RPE), Self-Assessment-Manikin (SAM), and Subjective Exercise Experience Scale (SEES). The results showed that the RPE scores were significantly higher in HIIE than in MICE. HIIE resulted in significantly lower pleasure scores using the SAM while arousal and dominance scores were significantly higher with HIIE compared to MICE. Positive well-being scores using the SEES were significantly lower with HIIE and both psychological distress and fatigue scores were significantly higher with HIIE. The results showed that affective responses with MICE were more positive than with HIIE, but no differences were found between normal and overweight/obese women. Based on these results, MICE may be a more acceptable exercise program for inactive women regardless of their weight status.

## 1. Introduction

Affective responses can be viewed as an umbrella term that encompasses related concepts, including core affect (e.g., hedonic valence (pleasure/displeasure) and arousal), emotions (e.g., anger, sorrow), and moods (e.g., happy, depressed) [1,2]. Affective responses to different exercise intensity have been previously studied. Ekkekakis et al. reported that non-overweight individuals describe more positive affective responses to exercise intensity below the ventilatory threshold (VT) and more negative affective responses when exercise intensity is higher than the VT. Affective responses can be more varied when the exercise intensity is close to the VT. Some individuals reported increases in pleasure and others decreases in pleasure [3].

The same research group expanded the study by recruiting participants who are normal weight, overweight and obese women. Obese women gave lower pleasure ratings to a one-bout exercise test, compared to normal- and over-weight women. The authors reported that the lower levels of pleasure experienced by obese women, compared to the non-obese women, could be due to their dramatically low levels of physical activity participation (i.e., physically inactive) [4]. Moreover, there was a significant decline in reported pleasure when the speed selected was 10% higher than what the participant would have self-selected [5]. Therefore, people usually report a higher level of enjoyment when performing exercise with moderate intensity or lower than the VT [2].

Affective responses to exercise represent an important opportunity to predict the future physical activity level [3]. Mattsson et al. reported that overweight/obese women may perceive higher exertion level than normal-weight women during a brisk walk session [6], and the perception of exhaustion can potentially affect their motivation to future physical activity. Inactive individuals who reported more positive affective responses to a moderate-intensity stimulus (64% maximal heart rate) during a single bout of exercise reported longer duration of physical activity at both 6 and 12 months later [7]. A recent systematic review showed that affective responses during moderate-intensity exercise were associated with the future physical activity behavior [8].

While lack of time has been reported to be the most common barrier to physical activity in Taiwan [9], high-intensity interval exercise (HIIE) has shown to be a time-efficient training strategy which provides similar effects of traditional moderate-intensity continuous exercise (MICE) in muscle oxidative capacity and exercise capacity [10]. However, it has been debated that HIIE may not be appropriate for physically inactive individuals as it may evoke a negative affect that may lead to an avoidant response to future exercise sessions [11]. Since HIIE contains multiple intervals of low-intensity activity, which differs from the traditional high-intensity continuous exercise (HICE) or MICE, individuals’ affective responses to HIIE may be different from that to traditional continuous exercise. In the review of Stork et al., the affective response was shown to be similar or more negative during HIIE compared to HICE and MICE [12]. Results from a recent meta-analysis indicated that HIIE was experienced as less pleasant than MICE but similar to HICE [2]. As far as we know, little research has investigated the affective response to HIIE in physically inactive women. In addition, no prior studies have compared affective response to HIIE between normal weight and overweight/obese women. Therefore, this study aimed to examine and compare affective responses to HIIE vs. MICE in inactive women with different weight status.

## 2. Materials and Methods

### 2.1. Participants

A total of 60 female participants were recruited through word of mouth or webpage from Kaohsiung Medical University, Taiwan, including 30 participants whose body mass index (BMI) indicated normal body weight (18.5–23.9 kg/m^2^) and 30 participants whose BMI indicated overweight/obese (24–34.9 kg/m^2^). All participants were informed about the purpose, procedures, and precautions of the study before they signed the Informed Consent Form approved by the Kaohsiung Medical University Institutional Review Board (KMUH-IRB-20140034).

To enroll in this study, participants had to be: (1) female between 20 and 40 years of age, which is the population with the lowest exercise participation rate in Taiwan [13] and (2) BMI between 18.5–34.9 kg/m^2^. Participants were excluded if they: (1) engaged in regular physical activity (i.e., moderate to high intensity exercise ≥ 3 times a week, ≥30 min a day for at least 6 months) [14]; (2) were morbidly obese (BMI ≥ 35 kg/m^2^) or had other physical contraindications to exercise (e.g., orthopedic problems, cardiac or metabolic disease) [14]; or (3) were pregnant, nursing or postmenopausal.

### 2.2. Procedures

All participants were asked to complete body composition measurement and the VO_2_max test during the first visit. Each participant then was scheduled for two additional visits for exercise sessions: high-intensity interval exercise (HIIE) and moderate-intensity continuous exercise (MICE). The order of the exercise sessions was randomized and counterbalanced, and the two exercise sessions were at least one day apart. Exercise sessions were scheduled between the 5th and 10th day post-menstruation for each participant, during the same time of a day and conducted by the same examiner. The temperature of the laboratory remained at 25 °C for all sessions. All participants were also instructed to: (1) have enough sleep the night before the test, (2) avoid caffeine consumption during the day of exercise session, and (3) abstain from food for at least 1 h prior to the exercise session.

During each exercise session, participants were asked to complete the SAM and their rated perceived exertion (RPE) assessments before each session, during the first 5 min, half-way through the exercise session, in the last 5 min, at the end of the exercise session, as well as at 5, 15 and 30 min post exercise session. The SEES was completed by the participants before and after the exercise session as well as at 5, 15 and 30 min post exercise session. Participants shared their preference for exercise mode when prompted after the completion of both exercise sessions. Since the SEES was difficult to measure during exercise, it was measured half-way through the exercise session.

### 2.3. Maximal Oxygen Consumption Test (VO_2_max)

The participants’ VO_2_max was measured using the Åstrand cycle test protocol [15]. The participants performed cycling on a stationary bike with 50 W of resistance for the first 2 min and then the resistance was incrementally increased by a 25 W increment of resistance for the following every 2 min. Participants were asked to pedal at 50 rpm, and reported their RPE [16] at the end of every 2-min stage. All participants were asked to wear heart rate monitors during the entire VO_2_max test. The test was terminated and the VO_2_max was determined when one of the following criteria was reached: (1) participants were unable to remain cycling at 50 rpm; (2) the heart rate failed to increase with increase in exercise intensity; or (3) the participant was fatigued and could not continue the testing.

### 2.4. Body Composition

Participants’ body weight, BMI, fat mass and body fat percentage were obtained by using a multifrequency bioelectrical impedance analyzer (Zeus 9.9 PLUS; Jawon Medical Co., Ltd., Kungsang Bukdo, Korea).

### 2.5. Self-Assessment Manikin (SAM)

Self-Assessment Manikin (SAM) is a non-verbal pictorial assessment technique that directly measures the pleasure, arousal, and dominance associated with a person’s affective reaction to an object or event [17]. SAM ranges from a smiling, happy figure to a frowning, unhappy figure when representing the pleasure dimension, and ranges from an excited, wide-eyed figure to a relaxed, sleepy figure for the arousal dimension. The dominance dimension represents changes in control with changes in the size of SAM: a large figure indicates maximum control in the situation. The participants can place an “x” over any of the five figures in each scale, or between any two figures, which results in a 9-point rating scale for each dimension.

### 2.6. Subjective Exercise Experience Scale (SEES)

The 3-factor SEES is a measure of global psychological responses to the exercise. The 3 factors comprise 4 items each: (1) positive well-being (PWB, including “Strong,” “Great,” “Positive” and “Terrific”); (2) psychological distress (PD, including “Crummy,” “Awful,” “Miserable” and “Discouraged”); (3) fatigue (FAT, including “Exhausted,” “Fatigued,” “Tired” and “Drained”). Participants were asked to score each item on a 7-point Likert scale with verbal anchors of “not at all” (1), “very much so” (7), and a midpoint anchor of “moderately so” (4).

### 2.7. Exercise Conditions 

#### 2.7.1. High Intensity Interval Exercise (HIIE) Condition

The exercise protocol used in the present study was adapted from previous published studies [14,15], and tested in 2 pilot studies with 20 volunteers. The exercise protocol consisted of a 3-min warm-up (30 W) and an 18-min main exercise session, in which participants cycled for 45 s at maximal intensity (Wmax) followed by 75 s of recovery at 50 W and then repeated this for 9 intervals. The participants were asked to maintain the cycling speed at 50 rpm through the entire training session.

#### 2.7.2. Moderate Intensity Continuous Exercise (MICE) Condition

Exercise intensity was 50% VO_2_ reserve for all participants [16]. The exercise duration was adjusted for each participant, so that the total energy expenditure of the MICE was the same as that of HIIE. The exercise protocol included a 3-min warm-up prior to the main exercise session.

### 2.8. Statistical Analyses

Data were analyzed using SPSS 19.0 (SPSS Inc, Chicago, IL, USA) for Windows. Descriptive statistics (mean ± SD) for continuous variables, such as age, BMI, percentage of body fat, heart rate, VO_2_max and maximal heart rate, were generated and compared between normal-weight and overweight/obese groups using the Student T-test. The differences in exercise modes between groups and at each time point were assessed using the three-way repeated measures analysis of variance (ANOVA). Significant F-tests were followed by post-hoc comparisons using the Bonferroni correction method. Estimates of effect size using the partial eta squared (η^2^) and Cohen’s d were reported to present the magnitude of the effect. The significance level (α level) was set at 0.05.

## 3. Results

### 3.1. Participant’s Characteristics

A total of 60 inactive women completed the study, including 30 women in the normal weight group and 30 women in the overweight/obese group (Table 1). There were no significant differences between the groups in age, height, resting heart rate (HRrest) and maximal heart rate (HRmax). Body weight, BMI, percentage of body fat and the maximal work rate during the exercise test were all greater in the overweight/obese group while VO_2_max was lower (all *p* < 0.05) (Table 1) The average duration for MICE was 24.75 ± 1.39 min in the normal-weight group and 21.73 ± 1.62 min in the overweight/obese group. All participants chose MICE when preference of MICE vs. HIIE was asked after the completion of both modes of exercise sessions.

### 3.2. Rating of Perceived Exertion (RPE)

In the overweight/obese group, the RPE was greater for HIIE, compared to MICE, during the exercise session, at the end of the exercise session, and at 5, 15, and 30 min post exercise session (all *p* < 0.03). Similar findings were found in the normal weight group, where the RPE was greater for HIIE than MICE during the exercise session, at the end of the exercise session, and at 5 and 15 min post exercise session (all *p* < 0.01). There was no significant difference between the groups in RPE pre-exercise or changes in the RPE during/post exercise session.

### 3.3. Self-Assessment Manikin (SAM)

#### 3.3.1. Pleasure

In the overweight/obese group, the score for pleasure was lower for HIIE compared to MICE during the exercise session, at the end of exercise session, and at 5 min post exercise session (all *p* < 0.01). Similar results were found in the normal weight group, where the score of pleasure was lower for HIIE than that for MICE during the exercise session, at the end of the exercise session, and at 5 and 15 min post exercise session (all *p* < 0.02). When measured at the pre-exercise session timepoint, the score of pleasure for HIIE was greater in the overweight/obese group than that in the normal weight group (*p* = 0.005). There was no significant difference between the groups in changes in the pleasure score during/post exercise session (Table 2).

#### 3.3.2. Arousal

In the overweight/obese group, the score for arousal was greater for HIIE compared to MICE during the exercise session, at the end of exercise session, and at 5 and 15 min post exercise session (all *p* < 0.05). Similar results were found in the normal weight group, where the score of arousal was greater for HIIE than MICE during the exercise session, at the end of the exercise session, and at 5 and 15 min post exercise session (all *p* < 0.02). There was no significant difference between the groups in the score of arousal pre-exercise or changes in the arousal score during/post exercise session (Table 2)

#### 3.3.3. Dominance

In the overweight/obese group, the score for dominance was greater for HIIE compared to MICE during the exercise session, at the end of exercise session, and at 5 and 15 min post exercise session (all *p* < 0.03). Similar results were found in the normal weight group, where the score of dominance was greater for HIIE than MICE during the exercise session, at the end of the exercise session, and at 30 min post exercise session (all *p* < 0.01). There was no significant difference between the groups in the score of dominance pre-exercise or changes in the dominance score during/post exercise session (Table 2).

### 3.4. Subjective Exercise Experience Scale (SEES)

#### 3.4.1. Positive Well-Being

In the overweight/obese group, the score for positive well-being was lower for HIIE compared to MICE at the end of exercise session, and at 5, 15 and 30 min post exercise session (all *p* < 0.05). Similar results were found in the normal weight group, where the score of positive well-being was lower for HIIE than MICE at the end of the exercise session and at 5, 15 and 30 min post exercise session (all *p* < 0.05). There was no significant difference between the groups in the score of positive well-being pre-exercise or the change in positive well-being score during/post exercise session (Table 2).

#### 3.4.2. Psychological Distress

In the overweight/obese group, the score for psychological distress was greater for HIIE compared to MICE at the end of exercise session, and at 5 and 30 min post exercise session (all *p* < 0.04). Similar results were found in the normal weight group, where the score of psychological distress was greater for HIIE than MICE at the end of the exercise session and at 5, 15 and 30 min post exercise session (all *p* < 0.02). There was no significant difference between the groups in the score of psychological distress pre-exercise or the change in psychological distress score during/post exercise session (Table 2).

#### 3.4.3. Fatigue

In the overweight/obese group, the score of fatigue was greater for HIIE compared to MICE at the end of exercise session and at 5, 15 and 30 min post exercise session (all *p* < 0.01). Similar results were found in the normal weight group, where the score of fatigue was greater for HIIE than MICE at the end of the exercise session and at 5, 15, and 30 min post exercise session (all *p* < 0.001). When compared between the groups, there was a significant difference in the change in fatigue score through the entire exercise course (*p* = 0.032); however, when separated by exercise modes, no significant difference was observed between the groups in either the HIIE or the MICE session. There was no significant difference between the groups in the score of fatigue pre-exercise (Table 2).

## 4. Discussion

The purpose of the present study was to investigate the affective responses to HIIE and MICE in inactive women, and furthermore, to compare the affective responses between normal-weight and overweight/obese inactive women in affective responses. The findings of this study indicated that the scores of RPE, SAM, and SEES showed more positive affective responses toward MICE, compared to HIIE; however, no difference was observed between normal-weight and overweight/obese women. In addition, all of the participants preferred MICE to HIIE.

Inactive women had higher RPE with HIIE than that with MICE, and this is consistent with the previous literature [18,19]. RPE was greater during higher intensity exercise, compared to lower intensity exercise in this population, despite the inclusion of a 75-s low-intensity period with the HIIE protocol. 

When affective responses were assessed using the SAM, pleasure was rated lower for HIIE than that for MICE, and remained lower even 15 min after the exercise session was terminated. Previous studies have found that the pleasure level was rated lower when participants performed higher intensity exercise (70% VO_2peak_ vs. 40% VO_2peak_) [20,21,22]. Since a higher pleasure level during an exercise session positively affects the future exercise compliance [23], it may be more appropriate to begin exercise at a lower intensity level in inactive women to ensure pleasure responses to exercise sessions and ultimately increase the possibility of exercise compliance in the future. The score for arousal was higher during HIIE than MICE, and persisted for 15 min post exercise session, aligning with previous research where individuals reported higher scores of arousal during higher intensity exercise [22]. The higher score in the dominance domain of the SAM scale indicates that the individual feels more dependent on others or feels dominated, while the lower score indicates the participant feels more able to control the current situation. Our results found that the dominance score with HIIE was significantly higher than that with MICE both during and post exercise. This suggests that these inactive women felt less able to control the situation during HIIE, and continued to have a lower sense of self-control even after 30 min of rest post exercise. Other research found higher dominance scores when the participants (24 healthy college women, aged 19–27 years) cycled at higher intensity (70% VO_2peak_) than at lower intensity (40% VO_2peak_) [21]. Compared to HIIE, participants reported higher scores for pleasure and arousal, and felt more able to control the exercise situation with MICE, suggesting that MICE may motivate inactive individuals to begin or adhere to an exercise program.

The score for positive well-being, measured by SEES, was lower after the HIIE session than that after the MICE session. Previous research found that the score of positive well-being was rated higher during a cycling session at 50% HRR, compared to that at 85% HRR in young women [18]. On the contrary, in active men, the score of positive well-being increased during a 20-min 70% HR_max_ cycling session while no change was found during a 40% HR_max_ cycling session. These studies imply that the perception and affective responses to exercise training modes vary between men and women and between active and inactive individuals [24]; thus, in addition to selection of an appropriate exercise mode, clinicians have to consider influences of sex when prescribing an exercise program for the inactive population.

Psychological distress was greater post HIIE session than that after the MICE session. These findings indicated that inactive women perceived a higher level of psychological distress after HIIE. Previous studies related to psychological distress showed mixed findings. One study showed that women reported greater psychological distress during a 80% VO_2peak_ exercise session than that during a 60% VO_2peak_ exercise session [25]. On the contrary, another study found that there was no difference in psychological distress between high- vs. low-intensity exercise in adults with regular physical activity [24]. The discrepancy might be explained by the prior level of exercise training. Individuals who regularly exercise may adapt more easily to high-intensity exercise than those without exercise experience or training. In other words, previous experiences and habits of exercise may affect the level of psychological distress to different exercise intensity.

Similar to the findings for psychological distress and RPE, the score of fatigue was higher post HIIE session than that after MICE. This finding complied with a previous study, where a higher level of fatigue presented during a 85% HRR cycling session than that during a 50% HRR cycling session in young women [18].

All of the participants chose MICE as their preferred mode of exercise compared to HIIE and rated higher positive affective responses to MICE than HIIE. Given the growing research evidence indicating that affective response to a single exercise session can affect the physical activity level and exercise compliance in the future [7], our results have significant implications for exercise prescription in inactive women. MICE, compared to HIIE, might be a preferred more pleasurable exercise intervention for inactive women regardless of their weight status. Prescribing MICE instead of HIIE might achieve greater initial adherence to regular exercise in this population and those positive exercise experiences may help to generate long-term exercise behavior change. 

These findings were found in women aged between 20 and 28 years; thus, the results may not be generalized to men and other age groups. In addition, the affective responses to exercise sessions, assessed by subjective questionnaires, could have been influenced by participants’ physical and psychological status during the day or by the interaction between the participant and the examiner. The exercise sessions in the present study were all scheduled between the 5th and 10th day post-menstruation for all the participants, at the same time in a day, and conducted by the same examiner to avoid the fluctuation of physical and psychological status. We only used one HIIE protocol in the present study; therefore, in the future, comparison of affective responses to multiple HIIE protocols (e.g., different work-to-rest ratios) is warranted. The conventional MICE seemed to be favored by all the participants in the present study; however, participants’ long-term compliance to a MICE program requires further examination.

## 5. Conclusions

Both normal-weight and overweight/obese inactive women reported higher levels of pleasure and positive affective responses to MICE, compared to HIIE. All participants chose MICE as their preferred mode of exercise for future training, suggesting that the affective response during a single bout of exercise session may be associated with the choice of future exercise sessions. Although HIIE is a time-efficient training strategy which provides health benefits similar to MICE, MICE appears to be a preferred approach producing more positive affective responses and possibly better adherence to exercise in the future for inactive women.

## Figures and Tables

**Table 1 ijerph-18-05393-t001:** Participant’s characteristics.

	Normal Weight (n = 30)	Overweight/Obese (n = 30)	*p*	CI		Cohen’s d
				Upper	Lower	
Age (years)	22.13 (2.08)	22.07 (1.95)	0.898	0.97	−1.11	0.03
Height (cm)	161.40 (4.92)	160.90 (5.20)	0.703	2.11	−3.11	0.1
Weight (kg)	52.77 (4.21)	72.76 (11.90)	0.000	24.93	11.52	2.24
BMI (kg/m^2^)	20.26 (1.42)	28.03 (3.75)	0.000	9.25	6.28	2.74
Body fat (%)	25.28 (2.92)	34.78 (3.97)	0.000	11.30	7.69	2.73
HRrest (bpm)	76.93 (9.51)	78.40 (9.09)	0.544	6.27	−3.34	0.16
HRmax (bpm)	178.83 (9.94)	175.97 (12.69)	0.334	3.02	−8.76	0.25
Wmax (W)	124.17 (16.72)	143.40 (19.32)	0.000	28.58	9.82	1.06
VO_2_max (mL/kg/min)	32.45 (3.31)	28.59 (3.23)	0.000	−2.16	−5.54	1.18

Data are mean (SD); CI = Confidence interval; BMI = Body Mass Index; HRrest = heart rate at rest; HRmax = heart rate maxima; Wmax = the maximal wattage of Åstrand cycle test; VO_2_max = maximal oxygen uptake.

**Table 2 ijerph-18-05393-t002:** “Pleasure, Arousal, Dominance, Positive well-being, Psychological distress and Fatigue” changes between the two groups during HIIE and MICE condition.

Time Phase	Normal Weight (n = 30)		*p*	Overweight (n = 30)		*p*	Time × Group × Condition *p*-Value	η^2^
	HIIE	MICE		HIIE	MICE			
Pleasure								
A	6.70 (1.32)	6.93 (1.44)	0.394	7.60 (1.07)	7.43 (1.17)	0.231	0.870	0.006
B	5.17 (1.39)	6.17 (1.12)	0.001	5.40 (1.38)	6.37 (1.22)	0.000		
C	5.33 (1.45)	6.40 (1.19)	0.001	5.47 (2.00)	6.57 (1.10)	0.003		
D	6.10 (1.16)	7.07 (1.08)	0.000	6.60 (1.52)	7.33 (1.03)	0.007		
E	6.67 (1.16)	7.20 (0.89)	0.018	7.30 (1.24)	7.50 (1.28)	0.297		
F	7.00 (1.08)	7.07 (1.48)	1.000	7.93 (1.08)	7.93 (1.17)	0.752		
Arousal								
A	2.97 (1.69)	2.87 (1.78)	0.682	3.03 (1.56)	2.80 (1.71)	0.379	0.844	0.007
B	5.37 (1.40)	4.53 (1.68)	0.017	5.43 (1.72)	4.23 (1.78)	0.000		
C	5.67 (1.69)	4.50 (1.78)	0.005	5.50 (1.94)	4.30 (1.62)	0.005		
D	4.03 (1.27)	3.17 (1.58)	0.001	4.00 (1.62)	3.20 (2.01)	0.050		
E	3.50 (1.48)	2.53 (1.59)	0.000	3.47 (1.57)	2.83 (1.74)	0.009		
F	2.40 (1.19)	2.30 (1.64)	0.655	2.57 (1.70)	2.37 (1.56)	0.246		
Dominance								
A	2.13 (1.25)	2.43 (1.63)	0.307	2.57 (1.46)	2.83 (1.82)	0.199	0.488	0.015
B	4.07 (1.70)	3.17 (1.34)	0.000	4.77 (1.68)	3.23 (1.38)	0.000		
C	4.57 (2.16)	3.07 (1.44)	0.000	4.70 (2.26)	3.30 (1.58)	0.000		
D	2.97 (1.54)	2.37 (1.52)	0.077	3.07 (1.31)	2.40 (1.30)	0.010		
E	2.53 (1.36)	2.10 (1.32)	0.068	2.77 (1.28)	2.23 (1.38)	0.021		
F	2.17 (1.29)	1.73 (1.11)	0.005	2.10 (1.16)	1.83 (1.23)	0.058		
Positive well-being								
A	19.37 (3.71)	19.53 (3.36)	0.756	20.63 (3.62)	20.13 (3.68)	0.231	0.913	0.004
C	14.90 (3.99)	19.10 (3.44)	0.000	16.40 (5.18)	20.07 (3.68)	0.000		
D	16.83 (3.73)	19.97 (3.22)	0.000	18.40 (4.70)	20.97 (3.66)	0.001		
E	18.40 (3.78)	20.37 (2.91)	0.001	19.70 (4.53)	21.40 (3.99)	0.006		
F	19.30 (3.40)	20.53 (2.68)	0.042	20.83 (4.40)	22.13 (4.13)	0.041		
Psychological Distress								
A	5.77 (1.94)	6.43 (3.36)	0.230	6.43 (2.79)	6.67 (3.14)	0.500	0.197	0.026
C	11.93 (5.25)	6.63 (3.03)	0.000	10.97 (5.20)	6.93 (3.51)	0.000		
D	9.17 (4.68)	5.43 (2.08)	0.000	8.60 (4.32)	6.23 (2.89)	0.000		
E	7.70 (3.91)	5.13 (2.01)	0.000	7.17 (3.41)	5.93 (3.23)	0.052		
F	6.13 (2.93)	5.20 (2.11)	0.018	6.50 (2.96)	5.60 (2.56)	0.039		
Fatigue								
A	7.13 (2.80)	7.63 (3.81)	0.480	8.07 (3.41)	8.37 (3.86)	0.637	0.948	0.003
C	20.10 (5.47)	12.93 (4.87)	0.000	18.60 (5.64)	10.70 (4.06)	0.000		
D	16.13 (5.32)	10.13 (4.94)	0.000	14.43 (4.93)	8.80 (3.95)	0.000		
E	12.37 (5.40)	7.87 (3.84)	0.000	12.37 (4.94)	7.53 (3.98)	0.000		
F	10.03 (4.77)	7.10 (3.78)	0.000	9.93 (4.16)	6.73 (4.38)	0.001		

Time phase: A = Pre-exercise, B = Mid-exercise, C = Post-exercise, D = 5 min after exercise (5th), E = 15 min after exercise (15th), F = 30 min after exercise (30th).

## Data Availability

The data are not publicly available due to privacy/ethical restrictions.

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
