# Peer review of "Affective Responses during High-Intensity Interval Exercise Compared with Moderate-Intensity Continuous Exercise in Inactive Women"

_ijerph, 2021, doi:10.3390/ijerph18105393_

Round 1

Reviewer 1 Report

  • The manuscript presents descriptives of emotional responses during two types of exercises, among overweight/obese and normal weight women but the authors do not test an explanation for the differences.
  • The result apply to one session of both types of exercises, while I think it would be really interesting to relate these findings to long-term exercise behavior and/or physical activity participation. 
  • The need for a comparison between overweight/obese and healthy weight women is not clear to me: as overweight/obese people tend to have lower levels of physical activity and lower fitness levels, it is logic that they perceive higher exertion levels than normal weight people.

Reviewer 2 Report

This study provides valuable information by comparing the emotional responses of women to exercises of different intensities. However, the following points are ambiguous and need to be corrected. 

  1. Although there is a word "sedentary women" in the title, it should provide a basis for referring to sedentary. Looking at lines 78-80, it can be seen that the subject has no exercise habits, but there seems to be no evidence that he is sedentary. If sedentary is used, it should be justified.
  2. Line 77 states that the subject's age requirement is 20 to 40 years old, and the reason and rationale for this decision should be given. 
  3. Line 78 states that the subject's BMI ranges from 18.5 to 35, and the reason and rationale for this decision should be given. 

Round 2

Reviewer 2 Report

I think the items I pointed out last time have improved and the quality of the entire treatise has improved. Therefore, we judge that it is possible to accept. 

Reviewer 3 Report

Thank you for your comments and the revised manuscript.